# Multi-Atlas Brain Network Classification through Consistency Distillation and Complementary Information Fusion

## Abstract

In the realm of neuroscience, identifying distinctive patterns associated with neurological disorders via brain networks is crucial. Resting-state functional magnetic resonance imaging (fMRI) serves as a primary tool for mapping these networks by correlating blood-oxygen-level-dependent (BOLD) signals across different brain regions, defined as regions of interest (ROIs). Constructing these brain networks involves using atlases to parcellate the brain into ROIs based on various hypotheses of brain division. However, there is no standard atlas for brain network classification, leading to limitations in detecting abnormalities in disorders. Some recent methods have proposed utilizing multiple atlases, but they neglect consistency across atlases and lack ROI-level information exchange. To tackle these limitations, we propose an Atlas-Integrated Distillation and Fusion network (AIDFusion) to improve brain network classification using fMRI data. AIDFusion addresses the challenge of utilizing multiple atlases by employing a disentangle Transformer to filter out inconsistent atlas-specific information and distill distinguishable connections across atlases. It also incorporates subject- and population-level consistency constraints to enhance cross-atlas consistency. Additionally, AIDFusion employs an inter-atlas message-passing mechanism to fuse complementary information across brain regions. Experimental results on four datasets of different diseases demonstrate the effectiveness and efficiency of AIDFusion compared to state-of-the-art methods. A case study illustrates AIDFusion extract patterns that are both interpretable and consistent with established neuroscience findings.

## 1 Introduction

In the field of neuroscience, a key objective is to identify distinctive patterns associated with neurological disorders (e.g., Alzheimer's, Parkinson's, and Autism) by the brain networks (Poldrack et al., 2009). Resting-state functional magnetic resonance imaging (fMRI) is widely employed among various neuroimaging techniques to characterize the connectivities among brain regions (Worsley et al., 2002). This results in brain networks where each node represents a specific brain region, referred to as a region of interest (ROI). Each edge indicates a pairwise correlation between the blood-oxygen-level-dependent (BOLD) signals of two ROIs (Smith et al., 2011), revealing the connectivity between brain regions and indicating which areas tend to be activated synchronously or exhibit correlated activities.

Brain networks model neurological systems as graphs, allowing the use of graph-based techniques to understand their roles and interactions (Kawahara et al., 2017; Lanciano et al., 2020; Wang et al., 2023). Constructing these brain networks involves using a specific atlas to parcellate the brain into ROIs. Various atlases based on different hypotheses of brain parcellation, such as anatomical and functional divisions, have been proposed to group similar fMRI regions and create ROIs (Tzourio-Mazoyer et al., 2002; Makris et al., 2006; Schaefer et al., 2018a). Although proper brain parcellation is essential for detecting abnormalities in neurodegenerative disorders (Long et al., 2022), there is no golden standard atlas for brain network classification. Relying on a single atlas for brain network analysis has two main drawbacks. First, some voxels may not be assigned to any specific ROI, potentially leading to the loss of important information. Second, each atlas is based on a different

parcellation hypothesis. The BOLD signal of an ROI is averaged from all voxels within it, possibly missing detailed information. To address these limitations, recent works have proposed using multiple atlases with different parcellation modes to enhance multi-atlas brain network analysis. Some methods (Chu et al., 2022; Mahler et al., 2023) independently encode brain networks from various atlases and then aggregate the graph representations as a late feature fusion scheme for the final prediction. Another approach (Lee et al., 2024) incorporates early feature fusion by incorporating multi-atlas information from the raw data and using the fused feature for representation learning. However, these methods (1) neglect the need of consistency across atlases, potentially leading to the under-utilization of cross-atlas information; and (2) lack ROI-level information exchange throughout the entire representation learning process, which could hinder the models' ability to discern complementary information across different atlases.

In this paper, we propose an Atlas-Integrated Distillation and Fusion network (AIDFusion) to address the aforementioned limitations by utilizing atlas-consistent information distillation and cross-atlas complementary information fusion. Specifically, AIDFusion introduces a disentangle Transformer to filter out inconsistent atlas-specific information and distill distinguishable connections across different atlases. Subject- and population-level consistency constraints are applied to enhance cross-atlas consistency. Furthermore, to facilitate the fusion of complementary information across ROIs in multi-atlas brain networks, AIDFusion employs an inter-atlas message-passing mechanism that leverages spatial information. In summary, our key contributions are:

- We propose a multi-atlas solution for brain network classification with fMRI data. AIDFusion takes full advantage of multi-atlas brain networks by enhanced atlas-consistent information distillation and intense fusion of cross-atlas complementary information.

- We evaluate AIDFusion on four resting-state fMRI brain network datasets for different neurological disorders. Our results demonstrate the superiority of AIDFusion over state-of-the-art baseline methods in terms of effectiveness and efficiency in brain network classification.

- We present a case study that underscores the intriguing, straightforward, and highly interpretable patterns extracted by our approach, aligning with domain knowledge found in neuroscience literature.

## 2 RELATED WORK

### 2.1 BRAIN NETWORK ANALYSIS WITH VARIOUS ATLASES

**Multi-atlas methods** introduce multiple brain atlases for each neuroimage, which can provide information that complements each other and offers ample details without being restricted by the parcellation mode. MGRL (Chu et al., 2022) pioneered the construction of multi-atlas brain networks using various atlases. It applied graph convolutional networks (GCNs) to learn multi-atlas representations and perform graph-level fusion for disease classification. METAFormer (Mahler et al., 2023) proposed a multi-atlas enhanced transformer approach with self-supervised pre-training for Autism spectrum disorder (ASD) classification. A graph-level late fusion was utilized to aggregate the representations of different atlases. Lee et al. (2024) employed a multi-atlas fusion approach that integrates early fusion on the raw feature to capture complex brain network patterns. STW-MHGCN (Liu et al., 2023a) constructs a spatial and temporal weighted hyper-connectivity network to fuse multi-atlas information, and Huang et al. (2020) adopt a voting strategy to integrate the classification results of different classifiers (each corresponding to a different atlas) for ASD diagnosis. CcSi-MHAHGEL (Wang et al., 2024) introduces a class-consistency and site-independence Multiview Hyperedge-Aware HyperGraph Embedding Learning framework to integrate brain networks constructed on multiple atlases in a multisite fMRI study. However, these studies did not consider the inherent consistency between atlases. Independently encoding multi-atlas brain networks without constraints might extract atlas-specific information, distracting from disease-related pattern modeling. Moreover, existing works only incorporate primitive early or late feature fusion between atlases. This absence of intermediate ROI-level interaction could hinder their models' ability to discern complementary information in each atlas. To the best of our knowledge, our work is the first to introduce information distillation with consistency constraints and employ intermediate ROI-level interaction for complementary information fusion. Note that in our work, multiple atlases are ap-

plied to preprocessed images for parcellation, meaning our method is based on a single template. The difference between multi-atlas methods and multi-template methods are discussed in Appendix A.1. The discussion about various atlases is included in Appendix A.2.

**Multi-modal and multi-resolution methods** also explore brain networks using various atlases. Research about multi-modal brain networks (Zhou et al., 2019; 2020; Zhu et al., 2022; Zhang et al., 2023; Qu et al., 2024) employed multiple modalities of neuroimaging data, including fMRI, Diffusion Tensor Imaging (DTI) and Positron Emission Tomography (PET), with various atlases to enhance brain network classification, as different modalities provide abundant information compared to a single modality. However, these multi-modal methods focus on fusing structural and functional connectivity information instead of trying to capture the whole picture of the single modality data. Another line of research (Zhang et al., 2020; Liu et al., 2021; 2023b; Wen et al., 2024) focuses on applying multi-resolution atlases to fMRI data to capture individual behavior across coarse-to-fine scales. However, the technical design of these approaches focuses on extracting information from both fine and coarse scales under the same parcellation mode. Although multi-modal and multi-resolution methods employ various atlases, they focus on different objectives from multi-atlas approaches, and the field of brain network analysis with multi-atlas is still in its infancy stage.

## 2.2 GRAPH NEURAL NETWORKS (GNNs)

In recent years, there has been a surge of interest in employing Graph Neural Networks (GNNs) for the analysis of brain networks. Ktena et al. (2017) utilized graph convolutional networks to learn similarities between pairs of brain networks (subjects). BrainNetCNN (Kawahara et al., 2017) introduced edge-to-edge, edge-to-node, and node-to-graph convolutional filters to harness the topological information within brain networks. MG2G (Xu et al., 2021) utilizes a two-stage method where it initially learns node representations using an unsupervised stochastic graph embedding model based on latent distributions, which are then used to train a classifier, allowing for the identification of significant ROIs with Alzheimer's disease (AD)-related effects. Zhang et al. (2022) incorporated both local ROI-GNN and global subject-GNN guided by non-imaging data. ContrastPool (Xu et al., 2024) introduced a dual attention mechanism to extract discriminative features across ROIs for subjects within the same group.

An alternative method for graph representation learning involves Transformer-based models (Vaswani et al., 2017), which adapt the attention mechanism to consider global information for each node and incorporate positional encoding to capture graph topological information. Graph Transformers have garnered significant attention due to their impressive performance in graph representation learning (Dwivedi & Bresson, 2020; Ying et al., 2021; Rampášek et al., 2022). A series of brain network Transformer methods have emerged for brain network analysis. One such method (Kan et al., 2022) applied Transformers to learn pairwise connection strengths among brain regions across individuals. Com-BrainTF (Bannadabhavi et al., 2023) uses a hierarchical local-global transformer for community-aware node embeddings, while GBT (Peng et al., 2024) employs an attention weight matrix approximation to focus on the most relevant components for improved graph representation. THC (Dai et al., 2023) introduced an interpretable Transformer-based model for joint hierarchical cluster identification and brain network classification. DART (Kan et al., 2023) utilized segmenting BOLD signals to generate dynamic brain networks and then incorporated them with static networks for representation learning. Most GNN- and Transformer-based methods for brain network analysis are designed for single-atlas thus may lead to a dependency on specific parcellation mode.

## 3 PRELIMINARIES

### 3.1 BRAIN NETWORK CONSTRUCTION

We introduce our method using two atlases, $a$ and $b$, for simplicity. It can easily generalize to more atlases. In this work, we use the datasets preprocessed and released by Xu et al. (2023). Each subject is characterized by two brain networks, represented by connectivity matrices $\boldsymbol{X}^a \in \mathbb{R}^{n_a \times n_a}$ and $\boldsymbol{X}^b \in \mathbb{R}^{n_b \times n_b}$. These matrices are derived using different atlases, which divide the whole brain into $n_a$ and $n_b$ ROIs respectively. In these matrices, each entry represent an edge in the brain network, which is calculated by Pearson's correlations between the region-averaged BOLD signals

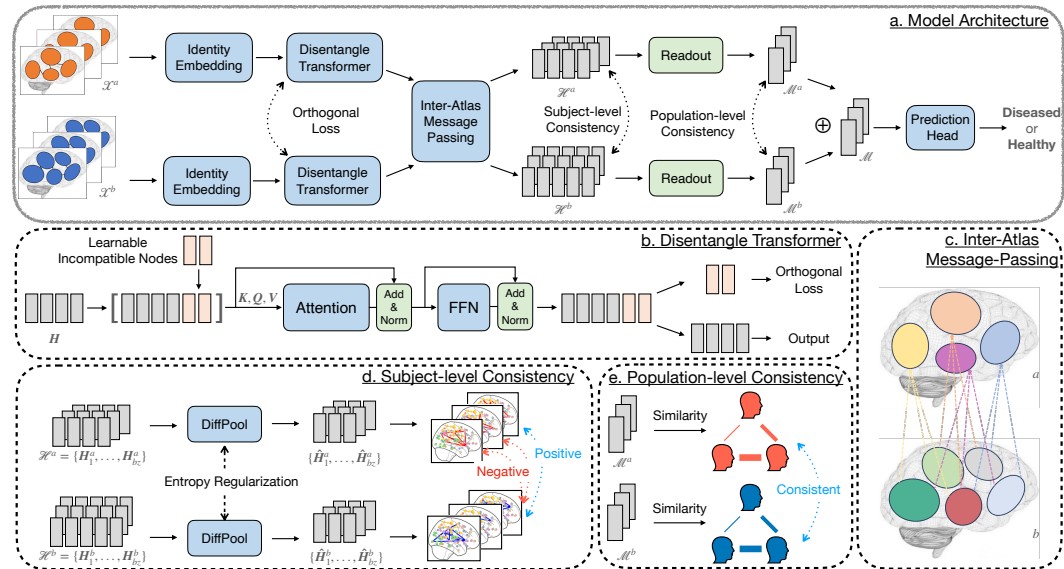

Figure 1: The framework of AIDFusion for multi-atlas brain network classification. The proposed framework includes three key components: Disentangle Transformer, Inter-Atlas Message-Passing, and Subject- and Population-level Consistency Constraint.

from pairs of ROIs. Thus these brain networks are weighed and fully-connected. Essentially, these brain networks capture functional relationships between different ROIs.

## 3.2 PROBLEM DEFINITION

Multi-atlas brain network classification aims to predict the distinct class of each subject by using various atlases for the same fMRI data. Given a dataset of labeled brain networks $\mathcal{D} = \{(\boldsymbol{X}^a, \boldsymbol{X}^b, y_X)\}$, where $y_X$ is the class label of brain networks $\boldsymbol{X}^a$ and $\boldsymbol{X}^b$, the problem of brain network classification is to learn a predictive function $f\colon (\boldsymbol{X}^a, \boldsymbol{X}^b) \to y_X$, which maps input brain networks to the groups they belong to, expecting that $f$ also works well on unseen brain networks. We summarize the notations used throughout the paper in Appendix B.

## 4 METHODOLOGY

In this section, we provide a detailed exposition of the design of our proposed Atlas-Integrated Distillation and Fusion network (AIDFusion), depicted in Figure 1. Two brain networks constructed with different atlases are separately processed in our model. In the following, we first introduce the disentangle Transformer with identity embedding to remove inconsistent atlas-specific information (Section 4.1). We then describe the inter-atlas message-passing for spatial-based intense fusion of cross-atlas (Section 4.2). Finally, we discuss our design of the losses that enforce atlas-consistent information distillation with domain considerations (Section 4.3).

### 4.1 DISENTANGLE TRANSFORMER WITH IDENTITY EMBEDDING

**Identity Embedding.** In graph Transformer models, positional embedding is commonly used to encode the topological information of the graph. However, designs like distance-based, centrality-based, and eigenvector-based positional embeddings (Li et al., 2020; Ying et al., 2021; Wang et al., 2022) are impractical for brain networks due to their high density (always fully connected). Correlation-based brain networks already contain sufficient positional information for ROIs, making general positional embeddings both costly and redundant. Instead, we propose a learnable identity embedding that adaptively learns a unique identity for each ROI, aligning nodes in different subjects that correspond to the same ROI in the same atlas. This embedding assigns the same identity to

nodes within the same ROI. As shown in Eq. (1), we introduce a parameter matrix $\boldsymbol{W}_{ID}$ to encode node identities alongside original node features $\boldsymbol{X}$, with $\mathrm{MLP}(\cdot)$ denoting a multilayer perceptron (MLP). More discussion about the identity embedding and positional embedding is provided in Appendix C.

$$\boldsymbol{H}_{ID} = \boldsymbol{X} + \mathrm{MLP}(\boldsymbol{X} + \boldsymbol{W}_{ID}). \tag{1}$$

**Disentangle Transformer.** Introducing learnable tokens in the input sequence of a Transformer has been a method used to capture global information. In natural language processing, Burtsev et al. (2020) first utilized a learnable $[CLS]$ token to improve machine translation tasks. In computer vision, Darcet et al. (2023) introduced register tokens to avoid recycling tokens from low-informative areas. Motivated by these prior works, we propose a disentangle Transformer to filter out inconsistent atlas-specific information by introducing incompatible nodes. We elaborate this module in Figure 1b. Specifically, given an identity-encoded graph feature matrix $\boldsymbol{H}_{ID} \in \mathbb{R}^{n \times d}$, where $n$ is the number of nodes and $d$ is the hidden dimension, we add $r$ learnable incompatible nodes $\boldsymbol{W}_{INC} \in \mathbb{R}^{r \times d}$ to the feature matrix:

$$\boldsymbol{H}' = \left[ \begin{array}{c} \boldsymbol{H}_{ID} \\ \boldsymbol{W}_{INC} \end{array} \right], \tag{2}$$

where $\left[ \begin{array}{c} \cdot \\ \cdot \end{array} \right]$ denotes the append operation. To enforce each incompatible node captures different information, we initialize them using the Gram-Schmidt process (Cheney & Kincaid, 2009) to ensure they are orthogonal to each other. Then the self-attention function (Vaswani et al., 2017) is applied to $\boldsymbol{H}' \in \mathbb{R}^{(n+r) \times d}$:

$$\mathrm{Attn}(\boldsymbol{H}') = \mathrm{norm}\left( \boldsymbol{H}' + \mathrm{softmax}\left( \frac{\boldsymbol{Q}\boldsymbol{K}^{\top}}{\sqrt{n+r}} \right) \boldsymbol{V} \right), \tag{3}$$

$$\boldsymbol{Q} = \boldsymbol{H}'\boldsymbol{W}_Q, \boldsymbol{K} = \boldsymbol{H}'\boldsymbol{W}_K, \boldsymbol{V} = \boldsymbol{H}'\boldsymbol{W}_V, \tag{4}$$

where $\boldsymbol{W}_Q, \boldsymbol{W}_K, \boldsymbol{W}_V \in \mathbb{R}^{d \times d}$ are parameter matrices and $\mathrm{norm}(\cdot)$ is a layer normalization.

In addition to the attention layer, a position-wise feed-forward network (FFN) with a layer normalization function is applied to each position to get the output node representations. The brain network of each atlas goes through a separate Disentangle Transformer. At the output of the disentangle Transformer, the incompatible nodes are discarded and only the ROI nodes are used.

**Orthogonal Loss.** As the brain networks derived from different atlases are based on the same fMRI data, we aim to ensure they contain similar information by filtering out inconsistent atlas-specific information. Therefore, we propose an orthogonal loss to enforce the representations of incompatible nodes to be orthogonal to each other across all atlases by minimizing their dot product:

$$\mathcal{L}_{orth} = \frac{1}{r} \sum \frac{||\boldsymbol{W}_{INC}^a \cdot \boldsymbol{W}_{INC}^b||}{||\boldsymbol{W}_{INC}^a|| \cdot ||\boldsymbol{W}_{INC}^b||}. \tag{5}$$

### 4.2 INTER-ATLAS MESSAGE-PASSING

The features at different atlases originate from totally different parcellation modes. Pulling those highly correlated features of two different atlases into a shared space allows their effective fusion. Existing literature on multi-atlas brain networks independently learns the representations of ROIs in each atlas without exchanging information across atlases (Chu et al., 2022; Lee et al., 2024). Additionally, the spatial relationship between ROIs in different atlases is neglected in these works. Our proposed AIDFusion enables inter-atlas message-passing between neighboring regions in different atlases by considering spatial information. Specifically, we use the spatial distance between the centroids of ROIs in different atlases to construct inter-atlas connections. As shown in Figure 1c, we utilize the $k$-nearest-neighbor ($k$NN) algorithm to connect each ROI to $k$ ROIs from the other atlas. The detailed description for the construction of inter-atlas adjacency matrix $\boldsymbol{A}^{ab}$ is included in Appendix D.1. Note that we only construct inter-atlas connections without considering intra-atlas connections since the information exchange within the same atlas has already proceeded in previous disentangle Transformer. Afterwards, an adjacency matrix $\boldsymbol{A}^{ab} \in \{0,1\}^{(n^a+n^b) \times (n^a+n^b)}$ is obtained and used for graph convolution (Kipf & Welling, 2016):

$$\mathrm{GCN}(\boldsymbol{A}^{ab}, \boldsymbol{H}^{ab}) = \sigma\left( \boldsymbol{D}^{-\frac{1}{2}} \boldsymbol{A}^{ab} \boldsymbol{D}^{-\frac{1}{2}} \boldsymbol{H}^{ab} \boldsymbol{W}_{GC} \right), \tag{6}$$

where $\sigma$ is the activation function (e.g., ReLU), $\boldsymbol{D}$ is the degree matrix of $\boldsymbol{A}^{ab}$, $\boldsymbol{H}^{ab} \in \mathbb{R}^{(n^a + n^b) \times d}$ is the combined node representation matrix for the two atlases, and $\boldsymbol{W}_{GC}$ is the learnable weight matrix of the GCN layer. An example of the adjacency matrix $\boldsymbol{A}^{ab}$ that used for inter-atlas message-passing is shown and discussed in Appendix D.2.

### 4.3 SUBJECT- AND POPULATION-LEVEL CONSISTENCY

**Subject-level Consistency.** To ensure the high-level consistency for the two brain networks from different atlases, we introduce a contrastive loss on the subject level. First, we apply DiffPool (Ying et al., 2018) to each atlas to capture higher-level patterns. The DiffPool contains two GCN layers. $\text{GCN}_{pool}$ is used to learn a cluster assignment matrix $\boldsymbol{S} \in \mathbb{R}^{n \times n'}$ as shown in Eq. (7). Herein, $n'$ is a pre-defined number of clusters controlled by a hyperparameter named the pooling ratio. The other $\text{GCN}_{emb}$ is used to obtain the embedded node feature matrix $\boldsymbol{Z} \in \mathbb{R}^{n \times d}$ as shown in Eq. (8). Both these two GCNs are defined similarly with Eq. (6). We sparsify the connectivity matrices $\boldsymbol{X}$ by keeping top 20% correlations and use them as the adjacency matrices $\boldsymbol{A}$ in these two GCNs, to avoid over-smoothing. The feature matrices of two atlas $\boldsymbol{H}$ are obtained from the output of inter-atlas message-passing.

$$\boldsymbol{S} = \text{softmax} \left( \text{GCN}_{pool}(\boldsymbol{A}, \boldsymbol{H}) \right). \tag{7}$$

$$\boldsymbol{Z} = \text{GCN}_{emb}(\boldsymbol{A}, \boldsymbol{H}). \tag{8}$$

Once we obtain the cluster assignment matrix $\boldsymbol{S}$ and the embedded node feature matrix $\boldsymbol{Z}$, we generate a new feature matrix $\hat{\boldsymbol{H}} \in \mathbb{R}^{n' \times d}$ by $\hat{\boldsymbol{H}} = \boldsymbol{S}^\top \boldsymbol{Z}$. This coarsening process can reduce the number of nodes to get higher-level node representations. To avoid GNN treating each ROI and each node cluster equally, we adopt an entropy regularization to the assignment matrices of each atlas:

$$\mathcal{L}_E = \frac{1}{n'} \sum_{j=1}^{n'} \left( \text{entropy}(\boldsymbol{S}^a[j, :]) + \text{entropy}(\boldsymbol{S}^b[j, :]) \right), \text{entropy}(\boldsymbol{p}) = -\sum_{j=1}^{n'} \boldsymbol{p}_j \log(\boldsymbol{p}_j). \tag{9}$$

We elaborate on the module of subject-level consistency in Figure 1d. Through two DiffPool layers, we produce high-quality representations for each atlas by extracting high-level node representations. Then we are able to apply a contrastive loss to them by considering representations from the same subject as positive pairs $\mathcal{P}^{pos} = \{(\hat{\boldsymbol{H}}_i^a, \hat{\boldsymbol{H}}_i^b)\} : i = 1, \ldots, bz\}$ and representations from different subjects as negative pairs $\mathcal{P}^{neg} = \{(\hat{\boldsymbol{H}}_i^a, \hat{\boldsymbol{H}}_{\neg i}^b) : i = 1, \ldots, bz\}$:

$$\mathcal{L}_{SC} = -\log \frac{\sum \sum_{(\hat{\boldsymbol{H}}_i^a, \hat{\boldsymbol{H}}_i^b) \in \mathcal{P}^{pos}} \exp(\text{sim}(\hat{\boldsymbol{H}}_i^a, \hat{\boldsymbol{H}}_i^b)/\tau)}{\sum \sum_{(\hat{\boldsymbol{H}}_i^a, \hat{\boldsymbol{H}}_{\neg i}^b) \in \mathcal{P}^{neg}} \exp(\text{sim}(\hat{\boldsymbol{H}}_i^a, \hat{\boldsymbol{H}}_{\neg i}^b)/\tau)}, \tag{10}$$

where $\tau$ is a temperature hyper-parameter to control the smoothness of the probability distribution (You et al., 2020), $bz$ is the batch size, and $\text{sim}(\cdot)$ denotes the cosine similarity function that is applied to the same row in the two matrices.

**Population-level Consistency.** The readout function $\boldsymbol{m} = \text{READOUT}(\boldsymbol{H})$ is an essential component of learning the graph-level representations $\boldsymbol{m} \in \mathbb{R}^d$ for brain network analysis (e.g., classification), which maps a set of learned node-level embeddings to a graph-level embedding. To further constrain the consistency for graph representations across different atlases, we introduce a mean squared error (MSE) loss on the population level. As shown in Figure 1e, a population graph $\boldsymbol{G}$ is constructed by computing the similarity of each two subjects' graph representations in the same atlas. The intuition here is we aim to maintain the relationship of subjects across atlases, instead of directly enforcing graph representations of two atlases to be the same. This constraint does not rely on the target label, as we only encourage the consistency of (dis)similarity across atlases rather than enforcing the similarity within each labeled group. Such loss is formulated as follows:

$$\mathcal{L}_{PC} = \frac{1}{bz} \sum (\boldsymbol{G}^a - \boldsymbol{G}^b)^2, \boldsymbol{G}[i, j] = \text{sim}(\boldsymbol{m}_i, \boldsymbol{m}_j), \boldsymbol{m}_i, \boldsymbol{m}_j \in \mathcal{M}, \tag{11}$$

where $\mathcal{M}$ is the set of graph representations in a batch.

**Total Loss.** The model is supervised by a commonly-used cross-entropy loss $\mathcal{L}_{cls}$ (Cox, 1958) for graph classification. The total loss is computed by:

$$\mathcal{L}_{total} = \mathcal{L}_{cls} + \lambda_1 * \mathcal{L}_{SC} + \lambda_2 * \mathcal{L}_{PC} + \lambda_3 * \mathcal{L}_E + \lambda_4 * \mathcal{L}_{orth}, \tag{12}$$

where $\lambda_1$, $\lambda_2$, $\lambda_3$ and $\lambda_4$ are trade-off hyperparameters for balancing different losses.

# 5 EXPERIMENTAL RESULTS

## 5.1 BRAIN NETWORK DATASETS

Table 1: Statistics and class information of brain network datasets used in this work.

| Dataset | Condition | Subject# | Class# | Class Name |
|---|---|---|---|---|
| ABIDE | Autism Spectrum Disorder | 1025 | 2 | {TC, ASD} |
| ADNI | Alzheimer's Disease | 1326 | 6 | {CN, SMC, MCI, EMCI, LMCI, AD} |
| PPMI | Parkinson's Disease | 209 | 4 | {NC, SWEDD, prodromal, PD} |
| Matai | Mild Traumatic Brain Injury (mTBI) | 60 | 2 | {pre-season, post-season} |

We use four brain network datasets from different data sources for various disorders, which are ABIDE (Craddock et al., 2013) for Autism (ASD), ADNI (Dadi et al., 2019) for AD, PPMI (Badea et al., 2017) for Parkinson's disease (PD), and Mātai for mild traumatic brain injury (mTBI) (Xu et al., 2023). Statistics of the brain network datasets are summarized in Table 1. The atlases we use are Schaefer (Schaefer et al., 2018a) and AAL (Tzourio-Mazoyer et al., 2002) with 100 and 116 ROIs, respectively. The detailed dataset description is provided in Appendix E while the implementation detail of our experiments is given in Appendix F.

## 5.2 BASELINE MODELS

We use 9 single-atlas methods and 6 multi-atlas methods as baselines to evaluate our proposed AIDFusion, including: (1) **Conventional machine learning (ML) models**: Logistic Regression (LR) and Support Vector Machine Classifier (SVM) from scikit-learn (Pedregosa et al., 2011). These models take the flattened upper-triangle connectivity matrix as vector input, instead of using the brain network. (2) **General-purposed GNNs**: GCN (Kipf & Welling, 2016) and Transformer (Vaswani et al., 2017). (3) **Single-Atlas Models tailored for brain networks**: Brain-NetCNN (Kawahara et al., 2017), MG2G (Xu et al., 2021), ContrastPool (Xu et al., 2024), BNT (Kan et al., 2022) and GBT (Peng et al., 2024). (4) **Multi-atlas models**: MultiLR (multi-atlas version of LR, concatenate the flatten feature of multiple atlases as input), MultiSVM (multi-atlas version of SVM, similar with MultiLR); MGRL (Chu et al., 2022); MGT (a multi-atlas version of Transformer with the same fusion mechanism as MGRL), METAFormer (Mahler et al., 2023) and LeeNet (Lee et al., 2024).

## 5.3 MAIN RESULTS

We report the classification accuracy on 4 brain network datasets over 10-fold cross-validation in Table 2. For certain diseases, the effectiveness/informativeness of different atlases is different. On Matai, all the 7 baselines attain better performance with AAL116 than with Schaefer100. On ADNI, 6 out of 7 baselines also perform better with AAL116. In contrast on ABIDE, 5 out of 7 baselines achieve better results with Schaefer100 than with AAL116. It demonstrates the importance of using multi-atlas for brain network analysis instead of relying on one specific atlas. Moreover, it is evident that the multi-atlas baselines with a simple late fusion mechanism (MGRL and MGT) outperform their respective single-atlas models (GCN and Transformer). This highlights the effectiveness of multi-atlas approaches in enhancing the performance of base models. However, conventional ML models (MultiLR and MultiSVM) fail to outperform their single-atlas versions in some cases, possibly due to their inability to effectively utilize multi-atlas features with simple concatenate fusion.

We can also observe that our proposed AIDFusion consistently outperforms not only all single-atlas methods but also state-of-the-art multi-atlas methods across all datasets. Specifically, AIDFusion achieves improvements over all multi-atlas methods on these four datasets by up to 9.76% ((75.00% - 68.33%) / 68.33% = 9.76% on Mātai). Our model gains larger performance improvement on small datasets (PPMI and Mātai) than on large datasets (ABIDE and ADNI), which meets the intuition that information utilization tends to be more critical in applications with smaller sample sizes. Moreover, the results demonstrate that AIDFusion tends to have lower standard deviations compared to other multi-atlas models, indicating the robustness of AIDFusion. This robustness is particularly desirable in medical applications where consistency and reliability are crucial.

Table 2: Graph Classification Results (Average Accuracy ± Standard Deviation) over 10-fold-CV. The first and second best results on each dataset are highlighted in **bold** and underline. The p-values of one-sided paired t-tests comparing our AIDFusion with the best multi-atlas baselines on the four datasets are 0.0116, 0.0380, 0.0830, and 0.0886, respectively.

| atlas | model | ABIDE | ADNI | PPMI | Mātai |
|---|---|---|---|---|---|
| Schaefer100 | LR | 64.81 ± 3.70 | 61.97 ± 4.24 | 56.48 ± 6.76 | 60.00 ± 20.00 |
| | SVM | 64.41 ± 5.09 | 61.52 ± 4.95 | 63.21 ± 8.62 | 56.67 ± 17.00 |
| | GCN | 60.19 ± 2.96 | 60.40 ± 4.89 | 54.02 ± 9.06 | 56.67 ± 17.00 |
| | Transformer | 59.90 ± 3.77 | 63.64 ± 2.61 | 59.33 ± 5.68 | 60.00 ± 20.00 |
| | BrainNetCNN | 65.75 ± 3.24 | 60.48 ± 3.29 | 57.33 ± 10.32 | 61.67 ± 13.33 |
| | MG2G | 64.41 ± 2.16 | 63.64 ± 5.10 | 55.45 ± 10.24 | 61.67 ± 19.79 |
| | ContrastPool | 65.01 ± 3.84 | 65.67 ± 6.64 | 64.00 ± 6.63 | 61.67 ± 13.02 |
| | BNT | 60.01 ± 5.33 | 66.39 ± 3.29 | 56.60 ± 10.82 | 60.00 ± 13.33 |
| | GBT | 61.76 ± 4.89 | 64.22 ± 2.67 | 59.07 ± 12.74 | 65.00 ± 18.92 |
| AAL116 | LR | 63.80 ± 3.00 | 64.06 ± 1.80 | 56.00 ± 7.79 | 66.67 ± 21.08 |
| | SVM | 65.72 ± 3.30 | 63.40 ± 1.90 | 64.12 ± 5.69 | 65.00 ± 20.34 |
| | GCN | 60.10 ± 5.74 | 61.24 ± 2.47 | 53.14 ± 8.82 | 65.00 ± 21.67 |
| | Transformer | 60.88 ± 4.39 | 63.27 ± 2.79 | 61.24 ± 7.22 | 63.33 ± 24.49 |
| | BrainNetCNN | 64.58 ± 6.29 | 62.52 ± 2.91 | 51.19 ± 9.24 | 66.67 ± 18.33 |
| | MG2G | 62.99 ± 4.01 | 64.41 ± 2.52 | 59.71 ± 9.11 | 70.00 ± 19.44 |
| | ContrastPool | 64.70 ± 3.26 | 66.33 ± 4.10 | 63.56 ± 7.90 | 65.00 ± 20.82 |
| | BNT | 58.95 ± 4.84 | 59.39 ± 4.44 | 54.14 ± 8.53 | 63.33 ± 24.49 |
| | GBT | 59.93 ± 3.82 | 58.35 ± 5.96 | 54.26 ± 10.58 | 60.00 ± 20.00 |
| Schaefer100 + AAL116 | MultiLR | 65.23 ± 5.13 | 64.99 ± 2.40 | 55.00 ± 6.25 | 56.67 ± 24.94 |
| | MultiSVM | 64.31 ± 5.24 | 65.21 ± 2.74 | 63.60 ± 7.66 | 58.33 ± 17.08 |
| | MGRL | 61.56 ± 4.90 | 62.74 ± 3.55 | 54.55 ± 10.67 | 68.33 ± 18.93 |
| | MGT | 63.32 ± 3.90 | 63.99 ± 4.34 | 62.14 ± 9.90 | 65.00 ± 22.91 |
| | METAFormer | 61.27 ± 4.05 | 66.52 ± 2.63 | 54.02 ± 8.81 | 61.67 ± 25.87 |
| | LeeNet | 61.28 ± 3.12 | 64.63 ± 1.34 | 60.74 ± 4.39 | 58.33 ± 17.08 |
| | AIDFusion (ours) | **66.35** ± 3.26 | **67.57** ± 2.04 | **66.00** ± 4.71 | **75.00** ± 13.44 |

Table 3: Results of more evaluation metrics (precision, recall, micro-F1, and ROC-AUC) on ABIDE dataset over 10-fold-CV. The best result is highlighted in **bold**.

| | Precision | Recall | micro-F1 | ROC-AUC |
|---|---|---|---|---|
| MGRL | 59.90 ± 6.42 | 60.03 ± 6.31 | 59.71 ± 5.10 | 61.39 ± 5.08 |
| MGT | 60.33 ± 4.78 | 69.29 ± 6.74 | 64.21 ± 3.62 | 63.60 ± 3.80 |
| METAFormer | 59.33 ± 4.05 | 61.91 ± 7.79 | 60.20 ± 4.26 | 61.31 ± 3.94 |
| LeeNet | **64.30** ± 5.24 | 43.04 ± 6.09 | 51.23 ± 4.74 | 60.44 ± 3.06 |
| AIDFusion (ours) | 62.25 ± 3.00 | **74.80** ± 4.38 | **67.90** ± 3.12 | **66.73** ± 3.25 |

In addition to accuracy, we also report other evaluation metrics, including precision, recall, micro-F1, and ROC-AUC, for all the multi-atlas deep models on the ABIDE dataset. As displayed in Table 3, AIDFusion performs the best across all these metrics except for precision. We observe that compared to other baselines, our AIDFusion can significantly improve recall without compromising precision. Moreover, in medical diagnostics, it is crucial to ensure that all individuals with a certain condition are correctly identified, even if it leads to some false positives. Missing a true positive (failing to diagnose a disease) can have severe consequences, while false positives can be further examined or retested. Therefore, models with higher recall rates, such as our AIDFusion, are more suitable for real-life medical auxiliary diagnosis.

Besides Schaefer100 and AAL116, we also conduct experiments with 3 atlases on ADNI by using HO48 (Makris et al., 2006), which is discussed in Appendix G. AIDFusion is a clear winner under all multi-atlas settings. The influence when using atlases with various resolutions is also discussed in Appendix G.

## 5.4 MODEL INTERPRETATION

In neurodegenerative disorder diagnosing, identifying salient ROIs/connections associated with predictions as potential biomarkers is crucial. In this study, we utilize attention scores from the Trans-

former layer to generate heat maps for brain networks to interpret our model. We visualize these attention maps using the Nilearn toolbox (Abraham et al., 2014). Figure 2 presents attention maps for two atlases, where higher attention values mean better classification potential for AD (from the ADNI dataset). We utilized 7 networks (Yeo et al., 2011) to assess the connections between our highlighted ROIs and major networks potentially involved with disorders. ROIs from the AAL that do not overlap with these seven networks are excluded from the heat maps. The top 10 ROIs with the highest attention values are displayed in the brain view. As depicted in the attention maps, attention maps of both Schaefer and AAL atlases identify common connections between the visual network (VIS) and the dorsal attention network (DAN), recognized as key connectivities in AD research (Brier et al., 2012; Agosta et al., 2012). Additionally, atlas-specific connections are highlighted. For example, the attention map of Schaefer atlas emphasizes connections within the default mode network (DMN) corresponding with the observations of Damoiseaux et al. (2012). Findings on the attention map on AAL are consistent with Agosta et al. (2012), showing that AD is associated with connectivities in VIS, especially in frontal networks. These findings suggest that AIDFusion effectively captures complementary information from different atlases. We also find some highlighted ROIs that diverge from conventional neuroscientific understanding. For example, connections between VIS and somatomotor network (SMN) have a high attention weight in AIDFusion on Schaefer atlas, which may imply AD is related to the function of defining the targets of actions and providing feedback for visual activation. This insight has not been identified by existing literature. Discussion of ASD (from the ABIDE dataset) is included in Appendix H.

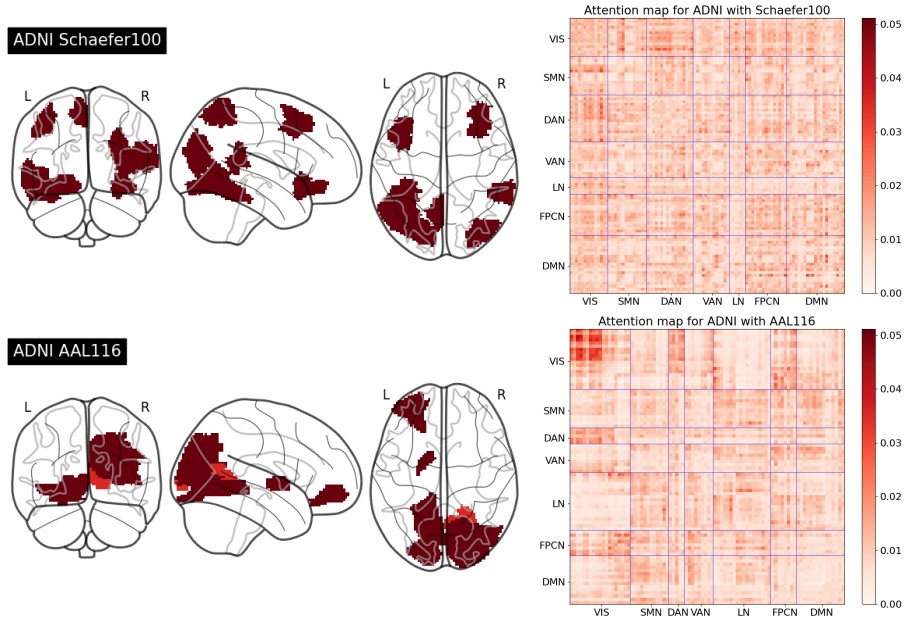

Figure 2: Visualization for attention maps on ADNI. VIS = visual network; SMN = somatomotor network; DAN = dorsal attention network; VAN = ventral attention network; LN = limbic network; FPCN = frontoparietal control network; DMN = default mode network.

## 5.5 ABLATION STUDY

To inspect the effect of the key components in AIDFusion, we conduct experiments by disabling each of them without modifying other settings. The results on ADNI dataset are reported in Table 4. For inter-atlas message-passing (denoted as "IA-MP" in the table), subject-level consistency and population-level consistency, we disable them by simply removing these modules. When disabling "Disen TF", we replace the disentangle Transformer and the identity embedding with a vanilla Transformer backbone (denoted as "TF" in the table). We further conduct experiments by disabling the identity embedding (shown as "Disen TF w/o ID"), and the orthogonal loss (shown as "Disen TF w/o $L_{orth}$") described in Section 4.1. When disabling all key components (the first row in the table), our model will degenerate to MGT in Table 2. The results demonstrate that AIDFusion with

all important modules enabled achieves the best performance. The component that affects the performance most is the population-level consistency. Besides, all variants of the proposed AIDFusion outperform the MGT baseline, demonstrating the effectiveness of our model design. We also conduct a case study of the role of the subject-/population-level consistency and incompatible nodes in the disentangle Transformer in Appendices I.1 and I.2, respectively. Further hyperparameter sensitivity analysis is provided in Appendix J.

Table 4: Ablation study on the key components of AIDFusion on ADNI, with the best result **bold**.

| Backbone | IA-MP | Subject-level Consistency | Population-level Consistency | acc ± std |
|---|---|---|---|---|
| TF | | | | 63.99 ± 4.34 |
| TF | ✓ | ✓ | ✓ | 66.82 ± 1.25 |
| Disen TF w/o ID | ✓ | ✓ | ✓ | 66.97 ± 1.35 |
| Disen TF w/o $\mathcal{L}_{orth}$ | ✓ | ✓ | ✓ | 66.06 ± 1.17 |
| Disen TF | | ✓ | ✓ | 66.58 ± 1.72 |
| Disen TF | ✓ | | ✓ | 66.37 ± 1.56 |
| Disen TF | ✓ | ✓ | | 65.91 ± 2.08 |
| Disen TF | ✓ | ✓ | ✓ | **67.57** ± 2.04 |

## 5.6 TIME EFFICIENCY

We conducted an experiment to compare the total runtime cost of AIDFusion with other multi-atlas baselines. Note that our method requires computing population-level consistency, which scales with the number of subjects rather than being a linear per-subject cost. The results, reported in Table 5, demonstrate that AIDFusion requires dramatically fewer epochs to converge, resulting in significantly less time spent on ABIDE, ADNI, and PPMI datasets. For the Mātai dataset, AIDFusion's time cost is still comparable with the other baselines. Besides, since AIDFusion does not contain any repeat layers as other baselines do, it has fewer parameters and thus results in higher efficiency. This showcases the efficiency of the proposed AIDFusion.

Table 5: Time efficiency analysis. Total time (h) was recorded with a single run (including training, validation, and test) with 10-fold CV.

| | ABIDE | | ADNI | | PPMI | | Mātai | | #Param |
|---|---|---|---|---|---|---|---|---|---|
| | Time (h) | #Epoch | Time (h) | #Epoch | Time (h) | #Epoch | Time (h) | #Epoch | |
| MGRL | 0.56 | 261.9 ± 0.7 | 0.91 | 262.7 ± 0.8 | 0.15 | 272.2 ± 8.0 | 0.09 | 291.2 ± 14.7 | 378k |
| MGT | 0.78 | 263.9 ± 2.4 | 0.89 | 108.2 ± 1.1 | 0.08 | 134.3 ± 10.9 | 0.18 | 266.4 ± 4.2 | 273k |
| METAFormer | 1.73 | 263.2 ± 1.5 | 1.47 | 268.3 ± 2.2 | 0.16 | 266.5 ± 4.7 | 0.12 | 270.4 ± 3.6 | 1886k |
| LeeNet | 1.56 | 200.0 ± 0.0 | 1.76 | 200.0 ± 0.0 | 0.19 | 200.0 ± 0.0 | 0.07 | 200.0 ± 0.0 | 526k |
| AIDFusion (ours) | 0.12 | 48.5 ± 19.0 | 0.26 | 64.6 ± 14.1 | 0.03 | 34.9 ± 12.0 | 0.10 | 119.5 ± 13.8 | 235k |

## 6 CONCLUSION

In this paper, we presented the Atlas-Integrated Distillation and Fusion network (AIDFusion), a novel approach to multi-atlas brain network classification. The disentangle Transformer mechanism, combined with inter-atlas message-passing and consistency constraints, effectively integrates complementary information across different atlases and ensures cross-atlas consistency at both the subject and population levels. Our extensive experiments on four fMRI datasets demonstrate that AIDFusion outperforms state-of-the-art methods in terms of classification accuracy and efficiency. Moreover, the patterns identified by AIDFusion align well with existing domain knowledge, showcasing the model's potential for providing interpretable insights into neurological disorders. Currently, our discussion of multi-atlas brain networks is restricted to 3 atlases, i.e., AAL, Schaefer, and HO. It is also worth extending the exploration to more atlases to find out which one benefits our model the most. Besides, experiments for atlases with various numbers of ROIs (Appendix F) indicate that AIDFusion performs better when the two atlases have a similar number of ROIs. How to utilize atlases with different resolutions remains a problem. Exploring datasets with more neuroimaging modalities and more atlases with different resolutions will provide insights into how atlas resolution affects the performance of multi-atlas methods. We plan to advance multi-scale brain network analysis by creating public benchmarks and leveraging inter-scale information to capture the brain's hierarchical organization more effectively.

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
