# OpenReview forum: "Multi-Atlas Brain Network Classification through Consistency Distillation and Complementary Information Fusion"
_ICLR.cc/2025/Conference — Submitted to ICLR 2025_

### Official Review · Reviewer_9YF5 · 2024-10-29

**Soundness:** 2
**Presentation:** 3
**Contribution:** 2
**Rating:** 6
**Confidence:** 5

**Summary:**

This paper proposes a method called AIDFusion, designed to improve brain network classification using rs-fMRI data.

The authors note that existing methods often rely on single atlases for classification, while approaches that utilize multiple atlases tend to overlook cross-atlas consistency and fail to facilitate information exchange at the ROI level. To address these issues, the authors first introduce a disentangled Transformer to learn atlas-level embeddings. They then propose an inter-atlas message-passing mechanism to fuse complementary information across brain regions. Additionally, subject-level and population-level consistency losses are employed to enhance cross-atlas coherence.

**Strengths:**

- Originality: This paper demonstrates some novelty. For instance, the concept of inter-atlas message-passing is interesting.
- Quality: The methodology is validated across four datasets representing different diseases, effectively showcasing its effectiveness and efficiency.
- Clarity: The paper is well-structured and easy to follow, enhancing comprehension for readers.
- Significance: This research offers new insights into brain network analysis within the field of neuroscience.

**Weaknesses:**

**1. There are too many hyperparameters to tune, which undermines the credibility of the results.**

For example, the training loss comprises five components, with four parameters requiring tuning. Additionally, hyperparameters from other modules of the method, such as the k value in KNN (lines 258-259), the keeping ratio (lines 278-279), and the temperature parameter (lines 302-303), also need to be set appropriately.

**2. The presence of numerous networks to learn complicates the training process, potentially leading to instability.**

This includes the transformers in lines 216-240 and three GCNs in lines 263-267 and 280-285.

**3. The motivations for some modules are unconvincing.**

For example, in lines 216-221 and 242-245, the authors mention introducing incompatible nodes and using orthogonal loss to filter out inconsistent atlas-specific information, citing the [CLS] token in NLP as motivation. However, the relationship is not adequately established, and the effect of these incompatible nodes is not clearly illustrated or justified.

**4. The comparisons are insufficient.**

Given that the atlas embedding learning is a transformer-based method, comparisons with additional transformer-based methods, such as [1], should be included. Additionally, other multi-atlas methods, like [2], should be considered. the manuscript may lack ablation studies that directly concatenate embeddings from single-atlas methods across different atlases similar in [2][3].

*[1] Kan, X., Dai, W., Cui, H., Zhang, Z., Guo, Y., & Yang, C. (2022). Brain network transformer. Advances in Neural Information Processing Systems, 35, 25586-25599.*

*[2] Wang, W., Xiao, L., Qu, G., Calhoun, V. D., Wang, Y. P., & Sun, X. (2024). Multiview hyperedge-aware hypergraph embedding learning for multisite, multiatlas fMRI based functional connectivity network analysis. Medical Image Analysis, 94, 103144.*

*[3] Zhao, B. W., You, Z. H., Wong, L., Zhang, P., Li, H. Y., & Wang, L. (2021). MGRL: Predicting Drug-Disease Associations Based on Multi-Graph Representation Learning. Frontiers in genetics, 12, 657182. https://doi.org/10.3389/fgene.2021.657182*

**Questions:**

Given the small sample sizes of rs-fMRI datasets, as stated in lines 325-330, how did you overcome the risk of overfitting during the training process and determine hyperparameters?

---

> ### Author Response · Authors · 2024-11-21
> **Response to Reviewer 9YF5**
>
> **[W1. Too many hyperparameters to tune.]**
>
> Thank you for pointing this out. We agree that our model includes a range of hyperparameters, but we want to clarify that not all of them require meticulous tuning. Here are the details:
>
> - The keeping ratio (sparsify threshold, related to adjacency matrix construction) is applied for GNN-based models and is fixed at 20% based on the dataset’s recommended setting in [1].
> - The trade-off hyperparameters $\lambda_4$ (orthogonal loss) and the temperature parameter $\tau$ (subject-level consistency loss) are quite stable  across datasets. We found that setting $\lambda_4 = 1.0$ and $\tau = 0.75$ works consistently well, allowing us to use these default values without further tuning when adapting to new datasets.
> - The key hyperparameters that require tuning are $\lambda_1$, $\lambda_2$, $\lambda_3$ (for the different loss components), and the $k$ value in kNN. We performed a grid search for these parameters.
>
> **[W2. Numerous networks to learn complicates the training process that potentially leading to instability.]**
>
> We appreciate this feedback. While AIDFusion involves multiple components, it actually has fewer model parameters compared to several state-of-the-art multi-atlas methods. Table 5 in our manuscript shows that our model requires fewer parameters. Additionally, our experiments demonstrate that AIDFusion converges significantly faster especially on large datasets, requiring fewer epochs compared to existing methods, as shown in Table 5. Besides, regarding model stability, we can observe that the standard deviation of AIDFusion is at the low end among all models (as shown in Table 2). Even when compared with LR and SVM (which much fewer model parameters), the std of AIDFusion still much lower. This demonstrates the stability of our proposed method.
>
> **[Q1. How to overcome the risk of overfitting.]** Overfitting is indeed a common challenge in brain network analysis due to the limited sample size. To prevent our model from overfitting, we use the early stopping criterion, i.e., we halve the learning rate when there is no further improvement on the validation loss during 25 epochs and stop the training once the learning rate is smaller than the minimum rate we set. And we also include dropout layers and L2 regularization in our model to prevent overfitting. This description is included in Appendix F.
>
> **[W3. Justification of the effect of incompatible nodes and orthogonal loss.]** Please refer to the general response.
>
> **[W4. The comparisons are insufficient.]**  Please refer to the general response.
>
> [1] Data-driven network neuroscience: On data collection and benchmark. NeurIPS 2023

---

### Official Review · Reviewer_3zUh · 2024-11-04

**Soundness:** 3
**Presentation:** 3
**Contribution:** 2
**Rating:** 5
**Confidence:** 4

**Summary:**

In this paper, a novel method named AIDFusion is proposed to learn unified and informative representations of brain functional networks derived from multiple brain atlases, which are expected to facilitate the brain network based classification. The method consists of several modules including disentangle transformer, inter-atlas message-passing, subject-level and population-level consistency to learn atlas-consistent information. The proposed method has been evaluated on multiple datasets to demonstrate its effectiveness.

**Strengths:**

1. The idea of learning unified representation from multiple brain networks is promising to improve classification performance.
2. Multiple techniques are proposed to facilitate the unified representation learning, e.g., disentangle transformer, inter-atlas message-passing, and multi-level consistency.

**Weaknesses:**

1. The brain networks from different atlases will contain atlas-consistent information and atlas-specific information, both types of information may be informative for classification. While the proposed method adopts several techniques to get enhanced atlas-consistent information, the atlas-specific information may not be effectively captured. Not sure if the proposed disentangle transformer may discard atlas-specific information that are useful for classification.
2. As demonstrate in the experimental results, the fusion of multiple arbitrary atlases may not improve classification performance. It is not clear and not verified what atlases should be fused.

**Questions:**

1. For the identity embedding in the disentangle transformer. It is not clear how the identity embedding was implemented. "... a parameter matrix W_ID to encode nodes within the same ROI", but it looks like one node corresponds to one brain ROI in the proposed method.
2. The brain networks from different atlases will contain atlas-consistent information shared across atlases and atlas-specific information, both types of information may be informative for classification. While the proposed method adopts several techniques to get enhanced atlas-consistent information, the atlas-specific information may not be effectively captured. Not sure if the proposed disentangle transformer may discard atlas-specific information that are useful for classification.
3. The Eq.(9) is not correct. I think i refers to ROI in atlas a or b, not cluster.
4. The AAL atlas contains both cerebral cortex, subcortical structures, and cerebellum regions, while the Schaefer atlas only covers cerebral cortex regions. Not sure if it is proper to learn consistent representations from brain networks with different brain coverages.
5. As demonstrate in the experimental results, the fusion of multiple arbitrary atlases may not improve classification performance. A principled way to identify what atlases should be fused is needed.
6. When applied to more than 2 atlases, will the orthogonal loss, inter-atlas message passing, and subject/population-level consistency be computed for each pair of atlases?

---

> ### Author Response · Authors · 2024-11-21
> **Response to Reviewer 3zUh**
>
> **[W1, Q2, Q4. Disentangle transformer may discard atlas-specific information that are useful for classification.]**
>
> Thank you for pointing this out. We aim to filter out atlas-specific information that is harmful to the downstream task. This is done in a soft way by using incompatible nodes in disentangle transformer. The atlas-specific information is not entirely filtered out as we can observe from Fig. 2 that different atlas still focuses on different ROI connections.  In our design, we have applied the consistency constraints primarily at a higher representation level rather than directly at the ROI or feature level to balance this trade-off. The orthogonal loss is only an auxiliary objective, ensuring that the disentangled representations are distinct but not dominant in driving the training process. Additionally, our case studies in Section 5.4 and Appendix H illustrate that AIDFusion focuses on diverse connections from different brain networks, demonstrating its capability to capture atlas-specific information.
>
> Regarding Q4, the learned consistent representations from brain networks with different coverages (e.g., AAL covering cerebellum regions, while Schaefer focuses on cerebral cortex) are obtained at a high-level feature space. This consistency aims to extract information that is disease-specific, even if the coverage differs. In our case studies, the attention weights show that the cerebellum ROIs receive lower attention, which aligns with existing neuroscience findings suggesting a lesser relevance of cerebellum regions in neurological disorder classification.
>
> **[W2, Q5. It is not clear what atlases should be fused.]** Thank you for this observation. We agree that determining the optimal set of atlases for fusion is an open research question. However, the main focus of our work is to propose a robust method for integrating multiple atlases rather than conducting an exhaustive search for the best atlas combination. Our experimental results show that in most cases using two atlases achieves superior results to using a single atlas with the same model.  Base on this observation, we further explore combining AAL116 with Schaefer at different number of ROIs, reported in Table 11 of Appendix G . Based on the results, we observe that combining two atlases with similar numbers of ROIs (e.g., Schaefer100 and AAL116) tends to yield better performance, which may be due to the balanced information content from both anatomical and functional perspectives. We acknowledge that this is a preliminary finding, and more extensive exploration is needed to determine the most effective atlas combinations. This is something we plan to investigate in future work.
>
> **[Q1. Not clear how the identity embedding was implemented.]** We apologize for the confusion. Yes, one node corresponds to one brain ROI. Please refer to our general response for a detailed explanation. In brief, the identity embedding is implemented using a parameter matrix $ W_{ID} $ that encodes the identity information of each ROI in different subjects with the same atlas. Each node is assigned an embedding vector based on its ROI identity, helping to differentiate nodes that belong to the same ROI across different subjects.
>
> **[Q3. Typo in Eq. (9).]** Thank you for raising this issue. We have modified $i$ to $j$ in Eq. (9) of our revision.
>
> **[Q6. Loss function when extended to more than 2 atlases.]** Yes, when AIDFusion is extended to handle more than two atlases, it computes the orthogonal loss, inter-atlas message passing, and both subject-level and population-level consistency for each pair of atlases. This pairwise computation ensures that the model can fully leverage the complementary information across all available atlases while maintaining consistency. We have clarified this in Appendix G of our revision.

---

> ### Author Response · Authors · 2024-12-03
> **[Gentle Reminder] Discussion period is closing soon**
>
> Dear Reviewer 3zUh,
>
> Thank you again for your time and valuable suggestions on our work! We understand you are busy. As the discussion period is closing soon, could you please take a look of our response above and let us know if our explanation addresses your concerns? We are more than happy to provide more details if it does not. And we would sincerely appreciate it if you could jointly consider our responses above when making the final evaluation of our work.
>
> Sincerely,
>
> Authors

---

### Official Review · Reviewer_CMqs · 2024-11-07

**Soundness:** 3
**Presentation:** 3
**Contribution:** 2
**Rating:** 5
**Confidence:** 3

**Summary:**

Resting-state functional MRI is often used to predict behavioral phenotypes by measuring correlations between disparate regions/parcels of the brain. There are many different parcellation conventions (“atlases” in this context) and downstream analyses are often quite dependent on the choice of atlas.

Submission 4706 presents a classification framework that integrates multiple different atlas conventions for a given subject. They propose a transformer that extracts complementary information and gain modest accuracy increases across four different datasets.

**Strengths:**

- The use of [transformer registers](https://arxiv.org/abs/2309.16588) (herein called “incompatible nodes”) for potentially filtering out incompatible information across atlases is a novel and interesting application.
- The use of spatial distances between regions of interest in the brain when constructing the message-passing framework is interesting and, to my limited knowledge, not commonly done.
- Generally clearly and straightforwardly presented.

**Weaknesses:**

I do not work in this specific subfield and would be happy to revisit my score and look forward to the discussion phase. I also did not read the appendix so please correct me if I missed something.

### 1. Unclear motivation for transformer-based approach on processed connectomes

The datasets in fMRI connectomics are (understandably) limited in sample size, ranging here from N=60 to N=1300.

However, it is unclear how the submission can adequately train transformers on 60 data samples. Without inductive bias (beyond permutation equivariance), transformers require several orders of magnitude higher training sets across the literature. I do not see any mention of pretraining either on larger datasets (e.g. UKBiobank) that would potentially enable few-shot finetuning.

Further, could the authors please elaborate on why they chose to work only with the processed connectomes instead of the high-dimensional raw data where there may be more potential for self-supervised pretraining?

### 2. Using only two atlases in experiments

The submission motivates itself by claiming that multiple atlases provide complementary information (which is a reasonable hypothesis) but its experiments only use two atlases (Schaefer100 and AAL116). Given that there is a wide range of potential atlasing procedures and many other atlases are often used, could the authors please clarify why two atlases were used in the experiments?

### 3. Unclear significance

The main comparative results (presented in Table 2) are largely well within each others’ error bars without clear significance.

Is there high inter-subject or inter-site variability s.t. standard deviations are inflated? Plotting per-subject performance as a supplemental figure should clarify this.

Further, could the authors please perform significance tests with corrections for multiple comparisons s.t. readers can assess whether these gains are meaningful?

### 4. Baseline choices and clarifications

#### 4.1. Iterative/conventional baselines

The only iterative baselines included for comparison are logistic regression and SVMs. It is not mentioned whether these methods were regularized in any form (e.g. LASSO) and, if so, whether the regularization hyperparameters were tuned at all. As far as I’m aware, practitioners [largely use LASSO-style methods for behavior prediction](https://db.humanconnectome.org/megatrawl/HCP820_MegaTrawl_April2016.pdf) and find that the regularization weight strongly affects the results.

Further, to my knowledge, deep nets and methods such as kernel regression are largely equivalent on this task when tested on much larger sample sizes ( [reference](https://pubmed.ncbi.nlm.nih.gov/31610298/) ).

As all results are well within each other's error bars (see point 3 above), could the authors please detail why only two non-DL methods were benchmarked and whether these methods were regularized and tuned?

#### 4.2 Transformer baselines

Several transformer-based approaches to fMRI classification are neither cited nor benchmarked against. For example,
- https://link.springer.com/chapter/10.1007/978-3-031-43993-3_28 (MICCAI’23)
- https://link.springer.com/chapter/10.1007/978-3-031-72390-2_14 (MICCAI’24)

Additionally, while not a one-to-one comparison to these papers use of static splits vs. the submission using 10-fold CV, these papers report much higher performance on ABIDE (high-70s vs the paper’s mid-60s accuracy).

Could the authors please address differences relative to these works, whether they could be used as baselines, and what could explain the major performance differences?

### 5. Missing ablations for key contributions

The paper has several moving parts, all of which are claimed as novel contributions. However, the core parts of Section 4.1 (the identity embedding, the “disentangle Transformer”, and the orthogonal loss) are not ablated. Please do so in the rebuttal or future versions.

### 6. Minor
These aspects do not affect my score.

- The paper claims that not all regions of the brain are covered by all atlases and thus integrating multiple atlases may have benefits. Please correct me if I’m wrong but don’t atlases such as Shen and Craddock cover most of the brain?
- Somewhat orthogonally, multiple atlas _harmonization_ can also be performed as with iterative methods like [CAROT](https://www.sciencedirect.com/science/article/am/pii/S136184152300124X). Could the authors please clarify the differences in motivation for their approach vs. atlas harmonization? I ask as much of the paper is motivated by the goal of integrating information across atlases and there are existing methods to do so.

**Questions:**

- How is transformer training feasible on tiny datasets with sample sizes of N=60 and N=1300 without any self supervised pretraining?
- Why work with processed connectomes instead of raw data for transformer training?
- Please perform significance testing with multiple comparisons adjustments.
- Please describe how the conventional baselines were tuned and whether they were regularized
- Please add ablations for the key contributions in 4.1

---

> ### Author Response · Authors · 2024-11-21
> **Response to Reviewer CMqs [1/2]**
>
> **[W1.1, Q1. Transformer trained on small datasets without any self-supervised pretraining.]**
>
> We appreciate your comment. As you noted, sample sizes for brain network analysis are indeed limited. We used the largest publicly available datasets for this domain. In fact, previous multi-atlas works such as MGRL (184 subjects), LeeNet (470 subjects), and METAFormer (884 subjects) utilized even smaller datasets than ours. For single-atlas methods like BNT, experiments were conducted on ABIDE with 1009 subjects, still fewer than in ours. We are also trying to collect and prepare more data, such as the mentioned UKBiobank. However, it involves different acquisition protocols and extremely time-consuming preprocessing (even 6-7 hours per subject). So we cannot add further data in this stage. We will try to include more data in our further work.
>
> We acknowledge the potential of self-supervised pretraining to address the challenges of limited training data. However, building a brain-specific pretraining model leveraging the unique characteristics of brain data is part of our ongoing research and lies outside the primary scope of this work. Our current focus is on **architecture design** for multi-atlas information fusion rather than pretraining strategies. Notably, our baseline METAFormer employs self-supervised pretraining but does not exhibit significant performance gains compared to our approach. This can be attributed to METAFormer’s reliance on late fusion, which lacks intermediate interaction across atlases. The observed results underscore the importance of a tailored architecture, such as AIDFusion, to effectively **extract and integrate information** from multiple atlases.
>
> **[W1.2, Q2. Use processed connectomes instead of raw data?]** Thank you for the insightful comment. We chose to work with processed connectomes because they effectively capture functional co-activation patterns between brain regions. These functional connectivity metrics have been well-established in neuroscience as reliable markers for neurological disorder classification [1]. Raw fMRI data often contain substantial noise, including motion artifacts and physiological fluctuations, which can obscure critical features. Besides, in the dataset paper we reference [2], the authors compare graph-based methods with time-series-based methods. Their results show that graph-based approaches, utilizing the connectivity matrix, consistently outperform time-series-based methods.  This justifies our decision to use processed connectomes, aligning with the existing body of literature.
>
> **[W2. Use of only two atlases in experiments.]** We selected AAL and Schaefer atlases because they are among the most widely adopted in the field, representing anatomical and functional parcellation approaches, respectively. Their detailed information of these two atlases are discussed in Appendix A.2. Additionally, we have tested a three-atlases case in Appendix G, on which AIDFusion also outperforms all baselines. We observed that all models tested on three atlases achieved poorer performance than the ones tested on Schaefer100 and AAL116. Therefore, we chose to conduct experiments on these two atlases in the main paper.
>
> **[W3, Q3. Unclear significance of results.]** Thank you for pointing this out. We performed statistical tests to assess the significance of our results. Specifically, one-sided paired t-tests between AIDFusion and the best multi-atlas baselines yielded p-values of 0.0116, 0.0380, 0.0830, and 0.0886 respectively on the four datasets. This indicates that our model significantly outperforms existing methods on the two large datasets ABIDE and ADNI. However, on small datasets of PPMI and Matai with large standard deviations in 10 folds, t-tests are highly sensitive to outliers existing in the 10-fold
> results, thus decreasing the t-statistic calculated and lowering
> the chance of rejecting the null hypothesis. We have included these p-values in the caption of Table 2 in the revised manuscript for transparency.

---

> ### Author Response · Authors · 2024-11-21
> **Response to Reviewer CMqs [2/2]**
>
> **[W4.1, Q4. Conventional baseline clarifications.]**
>
> We appreciate your detailed feedback. We selected SVM and LR as conventional baselines because they have been shown to perform robustly in prior studies and the original dataset papers [2]. We also followed the same hyperparameter-tuning protocol as in the original work. Specifically, we tuned LASSO, Ridge, and their combinations for LR, and applied L2 regularization for SVM. The full list of tuned parameters is provided in Appendix F of the revised manuscript.
>
> Additionally, differences in reported accuracies may arise due to variations in data splits, preprocessing, and cross-validation strategies. Note that using static splits may also incur higher accuracy because the hyperparameter-tuning could overfit on the specific split. Our brain network construction follows a standard pipeline and we implement all baselines under the same framework for fair comparison. Our baseline performance is comparable with the results reported in the original paper of the datasets [2].
>
> **[W4.2. Transformer baselines.]** Please refer to the general response.
>
> **[W5, Q5. More ablation studies.]** Please refer to the general response.
>
> **[W6.1. Coverage of brain regions by atlases.]** While many atlases provide broad coverage of brain regions, not all voxels are assigned to regions across different atlases. As noted in Appendix A.2, up to 33.3% of voxels may differ between atlases like AAL, Schaefer, and HO. Even for the Craddock atlas mentioned in the comment, 16.1% voxels in Craddock atlas are not included AAL atlas, while 16.0% voxels in AAL atlas are not included in Craddock atlas. This variability supports our motivation to integrate multiple atlases for a more comprehensive representation of the brain.
>
> **[W6.2. Difference with atlas harmonization methods.]** We appreciate your question. While CAROT and other harmonization methods primarily focus on aligning data across different atlases to ensure consistency, our approach simultaneously emphasizes both consistency and complementary information extraction. Specifically, our method incorporates the Subject- and Population-level Consistency Constraint at a higher representation level, an auxiliary orthogonal loss, and the Disentangle Transformer to capture consistency across data while also retaining atlas-specific information. Furthermore, through Inter-Atlas Message-Passing, we enhance the extraction of complementary features. This dual focus enables our model to retain task-specific variations across atlases that are often overlooked by traditional harmonization methods. Harmonization methods typically concentrate on achieving consistency across datasets, yet this often neglects the task-specific discriminative information that is crucial for improving performance in downstream tasks. In contrast, our framework is designed as an end-to-end classification system, which not only ensures consistency but also maximizes task relevance by allowing complementary features from different atlases to contribute to the final classification task.
>
> [1] Modern network science of neurological disorders. Nature Reviews Neuroscience 2014
>
> [2] Data-driven network neuroscience: On data collection and benchmark. NeurIPS 2023

---

> > ### Comment · Reviewer_CMqs · 2024-11-24
> >
> > Thank you for the thorough response. Given the limited time remaining (my apologies for the delayed response), I'll focus on major points alone.
> >
> > > **[W2. Use of only two atlases in experiments.] "We observed that all models tested on three atlases achieved poorer performance than the ones tested on Schaefer100 and AAL116."**
> >
> > Does this then not negate the premise of the paper?
> >
> > The claim is that the proposed transformer can learn to integrate information from across multiple atlases. If two atlases are sufficient for the considered tasks, adding more atlases should, at worst, retain the same performance, not decrease it. If by the proposed mechanisms, the transformer learns to aggregate relevant features across multiple atlases, it should learn to ignore the third atlas.
> >
> > > **[W1.1, Q1. Transformer trained on small datasets without any self-supervised pretraining.]**
> >
> > I understand that sample sizes are limited in brain studies and the associated challenges. To clarify, I'm asking how it is possible to train transformers *from random initialization* on datasets with 60 graphs (as in this paper) when all other subdomains of the transformer literature demonstrate that these networks require several orders of magnitude larger sample sizes to start showing benefits over either networks with relevant inductive biases or iterative methods.
> >
> > For example, Table 2 (Schaefer+AAL) shows the opposite trends from what we expect w.r.t. data efficiency:
> > - On datasets with higher sample sizes (ABIDE/ADNI), we see minor differences w.r.t. regularized iterative non-DL baselines such as logistic regression and SVMs.
> > - On datasets with smaller sample sizes (PPMI/Matai), we see larger differences w.r.t. regularized iterative non-DL baselines such as logistic regression and SVMs.
> >
> > It is in data-limited settings (e.g. N=60 as in this work) where we would expect non-DL baselines to outperform DL. I'm open to being wrong, could the authors please elaborate on this point?
> >
> > > **[W3, Q3. Unclear significance of results.] "one-sided paired t-tests between AIDFusion and the best multi-atlas baselines yielded p-values of 0.0116, 0.0380, 0.0830, and 0.0886 respectively"**
> >
> > If testing between the best and second best, why are these conducted on the best *multiatlas* method specifically? Looking at Table 2, it does look like the 2nd best method for 3 out of 4 datasets is a single atlas method.

---

> ### Author Response · Authors · 2024-11-26
>
> **[W2. Three Atlases Experiments]**
>
> We acknowledge that adding more atlases does not always improve performance, which is similar to the observations in multimodal learning where incorporating additional modalities may not consistently yield better results. Specifically, adding a third atlas might introduce noise or conflicting information that outweighs its potential benefits.
>
> Besides, to further verify our finding that our method will benefit more from atlases with a similar number of ROIs, we conduct additional experiments on the ABIDE dataset using three atlases (Schaefer100, AAL116, and BASC122 [3]). Results shown in the following table demonstrate that for five multi-atlas methods, three of them (MGRL, METAFormer, and AIDFusion) achieve better performance with three atlases compared to any two-atlas combinations. Importantly, our proposed AIDFusion still outperforms all baselines in these settings. This reinforces our claim that AIDFusion effectively integrates multi-atlas information while remaining robust to the inclusion of additional atlases. Detailed results and discussion have been added to Appendix G in the revised manuscript. The experiments for ADNI under the same setting are still running and we will update the result to the next version of our paper.
>
> |     Schaefer100    |     AAL116    |     BASC122    |     model         |     acc ± std       |
> |--------------------|---------------|----------------|-------------------|---------------------|
> |     √              |     √         |     |     MGRL          |     61.56 ± 4.90    |
> |     √              |     √         |     |       MGT           |     63.32 ± 3.90    |
> |     √              |     √         |     |     METAFormer    |     61.27 ± 4.05    |
> |     √              |     √         |     |      LeeNet        |     61.28 ± 3.12    |
> |     √              |     √         |     |      AIDFusion     |     **66.35** ± 3.26    |
> |    |     √         |     √          |     MGRL          |     60.80 ± 5.12    |
> |    |     √         |     √          |      MGT           |     58.49 ± 5.64    |
> |    |     √         |     √          |      METAFormer    |     62.92 ± 5.79    |
> |    |     √         |     √          |     LeeNet        |     60.01 ± 4.00     |
> |    |     √         |     √          |     AIDFusion     |     **65.97** ± 3.60    |
> |     √              |          |     √          |     MGRL          |     63.14 ± 4.17    |
> |     √              |          |     √          |     MGT           |     59.90 ± 3.46    |
> |     √              |          |     √          |   METAFormer    |     62.53 ± 3.78    |
> |     √              |          |     √          |    LeeNet        |     59.12 ± 4.20      |
> |     √              |          |     √          |    AIDFusion     |     **64.79** ± 2.80    |
> |     √              |     √         |     √          |     MGRL          |     63.62 ± 4.57    |
> |     √              |     √         |     √          |     MGT           |     62.84 ± 3.85    |
> |     √              |     √         |     √          |     METAFormer    |     65.80 ± 5.61    |
> |     √              |     √         |     √          |     LeeNet        |     60.19 ± 3.77    |
> |     √              |     √         |     √          |     AIDFusion     |     **66.65** ± 4.14    |
>
> **[W1.1, Q1. Data Efficiency]**  We appreciate the reviewer’s concerns about training transformers on limited datasets. fMRI data are inherently high-dimensional and suffer from a low signal-to-noise ratio due to factors such as cardiac and respiratory processes or scanner instability [4]. This poses challenges for training machine learning models, particularly on small datasets.  Transformers, while not immune to these issues, are better equipped to model the highly nonlinear nature of functional interactions in brain networks. However, for larger-scale datasets that are collected from multiple sites, such as ABIDE (17 sites) and ADNI (89 sites), site-specific noise can complicate training, leading to potential overfitting. This explains the observed trends: transformers may generalize better in smaller datasets such as Matai, where cross-site variability is less pronounced, while non-DL methods struggle with the data’s high dimensionality and complexity.
>
> **[W3, Q3. Significance Tests]**   We focused on comparing AIDFusion with the best multi-atlas baselines due to their alignment with our problem definition. Following the reviewer’s suggestion, we conducted one-sided paired t-tests between AIDFusion and the overall best baselines for each dataset. The revised tests yielded p-values of 0.0775 and 0.0380 for ABIDE and ADNI datasets, respectively.
>
> [3] Multi-level bootstrap analysis of stable clusters in resting-state fmri. NeuroImage 2010
>
> [4] Structural deep brain network mining. KDD 2018

---

> ### Comment · Reviewer_CMqs · 2024-12-02
>
> Thanks again for the detailed response. As some of my initial concerns (e.g., adding relevant ablations) have been addressed, I am raising the score to 5. It is primarily not higher as the reported gains over baselines do not seem to be robust on the largest reported datasets, please see the discussion on significance testing.
>
> [ Regarding the rebuttal discussion around data efficiency, for clarity, I was referring to transformers (by design) not having inductive biases (beyond permutation equivariance) and therefore having to learn them from large datasets, not the inter-site variability of neuroimaging datasets. It is not clear how this is learnable from datasets with N=60 from random initialization without pretraining. ]

---

> > ### Author Response · Authors · 2024-12-03
> >
> > **[Significance Testing on Large Datasets]**
> > Thank you for your thoughtful feedback and for raising the score. We acknowledge your concerns about the robustness of our model’s gains on larger datasets. However, we would like to highlight that most of the second-best methods are single-atlas approaches. Despite our use of a relatively simple architecture based on Transformers and GCNs, our model demonstrates significant improvements over other multi-atlas methods. This highlights its strong foundation and expandability. We believe that with further incorporation of more sophisticated components, AIDFusion has the potential to achieve even greater performance improvements.
> >
> > **[Data Efficiency and Transformer Inductive Biases]**
> > We appreciate your clarification regarding data efficiency and the challenge of Transformers' lack of inductive biases. Typically, Transformers require large datasets to learn these biases, posing a challenge in neuroimaging with limited sample sizes. To address this, our approach incorporates specific design choices:
> > 1. **Atlas-Specific Consistency**: By integrating identity embeddings and disentangling consistent and incompatible information across atlases, the model inherently captures domain-specific inductive biases.
> > 2. **Orthogonal Constraints**: These constraints encourage diverse and non-redundant feature learning, mitigating overfitting risks in small datasets.
> >
> > Additionally, unlike deeper Transformer architectures like ViT, which have extensive parameters and require large-scale datasets, our Transformer is intentionally shallow (1 layer) and narrow (hidden dimension = ~100) with a single attention head. This minimal parameterization allows effective learning even on small datasets.
> >
> > Nonetheless, we acknowledge the inherent challenges of training Transformers from scratch in data-constrained settings. As part of our future work, we are exploring brain-specific pretraining approaches, which we believe will further enhance model performance and generalizability.

---

### Official Review · Reviewer_VhSn · 2024-11-07

**Soundness:** 3
**Presentation:** 3
**Contribution:** 3
**Rating:** 8
**Confidence:** 3

**Summary:**

This paper aims to classify brain networks from fMRI BOLD ROIs using multiple (potentially conflicting) atlases. The authors resolve conflicts with a deep neural network (specifically a "disentangle transformer") alongside external prior information related to subject and population constraints.

To evaluate the efficacy of the proposed approach, the authors compare against several classical and current methods on four disparate datasets with a classification downstream task. Their method achieves the greatest quantitative results compared to all others in nearly all cases.

**Strengths:**

This approach is well-motivated, as there is no consensus on the number of atlases to use. Using a single atlas conforms to the biases induced that particular atlas, and using multiple atlases involves the use of novel methods which do not regularize consistency across multiple atlases. The proposed approach aims to achieve both consistency across multiple atlases as well as providing their model ROI-level "information exchange." Accomplishing these two aims required the development and use of several techniques, including the proposed "disentangle transformer" and "identity embedding", the use of an orthogonality loss on a component of the representations, a nearly-linear learnable mapping for inter-atlas message passing, a bespoke contrastive loss for subject-level consistency, and a similarity loss for "maintain the relationship of subjects across atlases."

This paper is well-motivated and fairly comprehensive with respect to the classification experiments. The contribution of the approach as an improvement for classification of brain networks is clear. I recommend this paper be accepted.

**Weaknesses:**

In addition to my recommendation for acceptance, I enumerate some concerns I have below:

1. The technical claims are not fully supported. The "identity embedding" is either mis-named or its explanation is unclear. The embedding is not identity; it seems instead to simply be a learnable embedding. It is unclear whether the parameters of the MLP in Eq. 1 are learnable. If so, what is the purpose behind W_ID?
2. The efficacy of the "disentangle Transformer" is not fully supported. How well are conflicting atlases disentangled? Was the orthogonal loss actually useful towards separating shared and conflicting information across atlases? An experiment to support the authors' intuition is missing.
3. The same is true for both subject- and population-level consistencies as well as message-passing. The explanation is sensible in text, but the actual efficacy of the proposed architecture and losses is not described in the experiments and results. The reader sees that the downstream classification is improved in light of the proposed components, but it is unknown whether these components accomplish what they are intended to.
4. Some figure and table captions are too terse. Figure1, table 1, table 3 could benefit from better description to help the reader understand their contents.
5. Table 5 experiment times include training, validation, and testing across multiple cross-validation folds. This is unusual; instead, reporting training+validation time and then testing time (for a single subject) separately would be more conventional. This guides future users of the proposed method towards how much time and compute should be budgeted for training the approach as well as what hardware is necessary to use the proposed method at scale in their own work.

**Questions:**

These questions are adapted / copied from the weaknesses listed above:

1. It is unclear whether the parameters of the MLP in Eq. 1 are learnable. If so, what is the purpose behind W_ID?
2. The efficacy of the "disentangle Transformer" is not fully supported. How well are conflicting atlases disentangled? Was the orthogonal loss actually useful towards separating shared and conflicting information across atlases? An experiment to support the authors' intuition is missing.
3. The same is true for both subject- and population-level consistencies as well as message-passing. The explanation is sensible in text, but the actual efficacy of the proposed architecture and losses is not described in the experiments and results. The reader sees that the downstream classification is improved in light of the proposed components, but it is unknown whether these components accomplish what they are intended to. What evidence do the authors have towards these claims?

---

> ### Author Response · Authors · 2024-11-21
> **Response to Reviewer VhSn**
>
> **[W1, Q1.C larification of identity embedding.]** Please refer to the general response.
>
> **[W2, Q2. The efficacy of the disentangle Transformer is not fully supported.]** Please refer to the general response.
>
> **[W3, Q3. The actual efficacy of the proposed architecture and losses is not described.]**
>
> We appreciate the reviewer’s insightful feedback on the subject- and population-level consistency constraints in our AIDFusion model. To address this, we conducted an in-depth analysis and visualized the impact of these losses.
>
> 1. Subject-level Consistency Loss: We visualized the difference matrices of hidden feature representations $\hat{\boldsymbol{H}}$ for two different brain atlases (Schaefer100 and AAL116). The results demonstrate that the subject-level consistency loss significantly reduces the discrepancies between the hidden features learned from different atlases, thereby enhancing the stability of the model's representations. This alignment indicates that AIDFusion captures shared patterns across atlases effectively, contributing to improved generalization.
>
> 2. Population-level Consistency Loss: We analyzed the difference in similarity matrices $\boldsymbol{G}$, which reflects pairwise similarities of subjects across atlases. Without the population-level consistency loss, substantial discrepancies were observed, potentially confusing the model. In contrast, applying this loss resulted in better-aligned similarity matrices, ensuring consistency in graph-level representations across different views. This leads to more robust predictions, even when faced with variations in brain network atlases.
>
> Overall, these findings support the importance of both consistency constraints in aligning multi-atlas representations, improving model robustness, and enhancing performance in downstream tasks. We have added these visualizations and detailed explanations to Appendix I.1 of our revision for clarity.
>
> **[W4. Some figure and table captions are too terse.]** Thank you for pointing this out. We have added more details on the captions of Figure 1, Table 1 and Table 3 to make them more informative in the revision.
>
> **[W5. Report testing time for each subject.]** Thank you for this valuable feedback. We understand that reporting separate training, validation, and testing times is more conventional. However, due to the small size of the datasets and the relatively small number of parameters in our model, the testing time for each subject is extremely short and does not significantly impact the overall computational budget. Additionally, our method requires computing population-level consistency, which scales with the number of subjects rather than being a linear per-subject cost. For this reason, we reported the overall time cost rather than averaging it by number of subjects.

---

> > ### Comment · Reviewer_VhSn · 2024-11-25
> >
> > Thank you for your response and for addressing my concerns. Your comment that the method requires computing population-level consistency each time is actually quite important I think. This could be emphasized in the final manuscript to prevent misunderstandings, but is fine as-is. My rating remains at "accept".

---

> > > ### Author Response · Authors · 2024-11-26
> > > **Thank you**
> > >
> > > Thank you for your constructive feedback and positive recommendation. As your suggestion, we have added the explanation about the population-level consistency computation in Section 5.6 of our revision to prevent misunderstandings.

---

### Author Response · Authors · 2024-11-21
**General Response [2/2]**

**[More ablation study @CMqs, 9YF5]** Thank you for the suggestion. We have expanded our ablation studies in the revised manuscript (Table 4). Specifically, we tested our model without the disentangle Transformer (labeled "TF"), without the identity embedding ("Disen TF w/o ID"), and without the orthogonal loss ("Disen TF w/o $L_{orth}$"). The results show a consistent drop in performance when any of these components is removed, highlighting their importance in achieving the superior performance of AIDFusion.

|     backbone                 |     IA-MP    |     SC    |     PC    |     adni            |
|------------------------------|--------------|-----------|-----------|---------------------|
|     TF                       |              |           |           |     63.99 ± 4.34    |
|     TF                       |     √        |     √     |     √     |     66.82 ± 1.25    |
|     Disen TF w/o ID          |     √        |     √     |     √     |     66.97 ± 1.35    |
|     Disen TF w/o   $L_{orth}$    |     √        |     √     |     √     |     66.06 ± 1.17    |
|     Disen TF                 |              |     √     |     √     |     66.58 ± 1.72    |
|     Disen TF                 |     √        |           |     √     |     66.37 ± 1.56    |
|     Disen TF                 |     √        |     √     |           |     65.91 ± 2.08    |
|     Disen TF                 |     √        |     √     |     √     |     **67.57** ± 2.04    |

[1] Community-Aware Transformer for Autism Prediction in fMRI Connectome. MICCAI 2023

[2] GBT: Geometric-Oriented Brain Transformer for Autism Diagnosis. MICCAI 2024

[3] Brain Network Transformer. NeurIPS 2022

[4] Multiview hyperedge-aware hypergraph embedding learning for multisite, multiatlas fMRI based functional connectivity network analysis. MIA 2024

---

### Author Response · Authors · 2024-11-21
**General Response [1/2]**

We appreciate the reviewers’ constructive feedback and positive remarks on the paper. We are grateful for the recognition of our work being **well-motivated and promising** (VhSn, 3zUh), **well-structured and easy to understand** (CMqs, 9YF5), **novel and interesting application** of transformer registers (incompatible nodes) (CMqs), and having **comprehensive experimental results** (VhSn) that **effectively demonstrate its effectiveness** (9YF5), while **offering new insights** to the field of neuroscience (9YF5).

In this rebuttal, we have addressed the reviewers' main concerns and provided additional experiments and clarifications. Revisions in our manuscript are highlighted in blue. We are open to further discussions if there are any unresolved concerns or additional questions.

**[Clarify identity embedding @VhSn, 3zUh]** We appreciate the request for clarification regarding the identity embedding. The identity embedding is designed to incorporate the anatomical consistency of brain networks across subjects with the same atlas. Specifically, brain networks constructed with the same atlas share the same ROI definitions. For instance, the first node in any brain network constructed with the AAL atlas corresponds to the same ROI. Thus, we use the identity embedding $W_{ID}[1, :]$ to represent this specific ROI and add it to the corresponding node feature $X[1, :]$. This design serves as a positional embedding mechanism, similar to techniques used in graph Transformers, where the learnable MLP parameters help incorporate this positional information effectively. We have clarified this in Appendix C of our revision.

**[More evidence for the disentangle Transformer @VhSn, 9YF5]** The disentangle Transformer aims to capture both atlas-specific information and filter out incompatible features across atlases. Drawing inspiration from the use of additional tokens in the NLP and CV domains for extracting global information, we introduced the incompatible nodes with an orthogonal constraint to disentangle shared and conflicting features. Discussion in Section 5.4 and Appendix H demonstrates the ability of AIDFusion to extract atlas-consistent information. To further justify this design, we conducted a case study by visualizing the attention maps of AIDFusion with and without incompatible nodes. The results show that without incompatible nodes, the attention maps are highly imbalanced, with much stronger attention on the Schaefer atlas compared to the AAL atlas. Furthermore, the attention map of AAL without incompatible nodes exhibits over-smoothing, failing to highlight distinguishable network connections. This indicates that the model struggles to extract informative features when inconsistent atlas-specific information is not filtered out. These findings support the necessity of incompatible nodes, and we have included the visualization and discussion in Appendix I.2.

**[More baseline comparison @CMqs, 9YF5]** Regarding Transformer-based baselines, BNT [3] applied Transformers to learn pairwise connection strengths among brain regions across individuals; Com-BrainTF [1] uses a hierarchical local-global transformer for community-aware node embeddings; GBT [2] employs an attention weight matrix approximation to focus on the most relevant components for improved graph representation. For multi-atlas related work, CcSi-MHAHGEL [4] introduces a class-consistency and site-independence Multiview Hyperedge-Aware HyperGraph Embedding Learning framework to integrate brain networks constructed on multiple atlases in a multisite fMRI study. We add all these methods into discussion in Section 2 of our revision. We also include comparisons with GBT [2] and BNT [3] in our updated results (Table 2). As shown in the following table, AIDFusion continues to achieve superior performance. Com-BrainTF [1] and CcSi-MHAHGEL [4] are not included because they rely on additional information (community and site distribution) that are not applicable to our setting.

|     atlas               |     model    | ABIDE               | ADNI                | PPMI                 | Matai                |
|--------------------------------|--------------|---------------------|---------------------|----------------------|----------------------|
|         Schaefer100            |     BNT      |     60.01 ± 5.33    |     66.39 ± 3.29    |     56.60 ± 10.82    |     60.00 ± 13.33    |
|                                |     GBT      |     61.76 ± 4.89    |     64.22 ± 2.67    |     59.07 ± 12.74    |     65.00 ± 18.92    |
|         AAL116                 |     BNT      |     58.95 ± 4.84    |     59.39 ± 4.44    |     54.14 ± 8.53     |     63.33 ± 24.49    |
|                                |     GBT      |     59.93 ± 3.82    |     58.35 ± 5.96    |     54.26 ± 10.58    |     60.00 ± 20.00    |
|         Schaefer100 +   AAL116 | AIDFusion    |     **66.35** ± 3.26    |     **67.57** ± 2.04    |     **66.00** ± 4.71     |     **75.00** ± 13.44    |

---

### Meta-Review · Area_Chair_Stxz · 2024-12-21

**Metareview:**

This paper introduces a transformer for fMRI data (presumably only resting state correlation matrices). They present classification task accuracy on four datasets (ABIDE, ADNI, PPMI, and Matai).

Reviewer responses are mixed, though discussion has improved scores. There remain unanswered questions however. Among these, specifically concerning are the high standard error; after rebuttal, there are now reportedly low p-values, but these are paired. This appears unanswered, as raised by `CMqs` and `3zUh`, and inconsistent with literature results, as raised by `9YF5`. Reviewer `9YF5` also raises questions about the number of hyper-parameters that may be tuned. Even if all other claims are believed at face value, this is problematic.

There are positive aspects to this submission and innovations in their work. However, it is below acceptance quality, and experimental concerns are unacceptable in general.

Beyond this, I find that the motivation is unclear. Is atlas fusion, embedding, or disambiguation the science in question? If so, we should see experiments showing that this model consistently identifies "correct" atlases. This is both a question at the identity embedding stage and at the "disentangle transformer" stage, both of which remain untested experimentally (with respect to identification properties). If the purpose is not better atlases (or identifying a "true" atlas), then pure classification remains the motivating factor, which is weak.

I encourage the authors to incorporate the concerns of Reviewer `CMqs` in particular. I find that the contributions of this manuscript have merit, but remain unpolished and leave questions unexplored in the science of brains and fMRI. Seeing as the transformer contributions are low (which is admittedly not the purpose of the paper; it is an "application" paper), for the above reasons I recommend rejection of this submission.

**Additional Comments On Reviewer Discussion:**

Overall the reviewers have highlighted several strong points, but have also opined on specific pieces that can be improved.
* Statistical and experimental problems (as detailed above).
* Overall writing problems ("Identity Embedding" and "Disentangle Transformer" break with the usual cases of terminology)
* Inclusion of large baseline datasets. ABCD and HCP (YA and Aging) are high quality imaging datasets which are also available.

I feel that the manuscript has improved through this review process. Inclusion of the ablation test in particular is useful.

---

### Decision · Program_Chairs · 2025-01-22

Reject